# A High-Precision and Low-Cost Broadband LEO 3-Satellite Alternate Switching Ranging/INS Integrated Navigation and Positioning Algorithm

Lvyang Ye [1] , Ning Gao [1], Yikang Yang [1,*] and Xue Li [2]

1   School of Electronic and Information Engineering, Xi'an Jiao Tong University (XJTU), Xi'an 710049, China
2   Communication, Telemetry and Command Center, Chongqing University, Chongqing 400044, China
*   Correspondence: yangyk74@mail.xjtu.edu.cn

**Abstract:** To solve the problem of location services in harsh environments, we propose an integrated navigation algorithm based on broadband low-earth-orbit (LEO) satellite communication and navigation integration with 3-satellite alternate switch ranging. First, we describe the algorithm principle and processing flow in detail; next, we analyze and model the ranging error source and propose a combined multipath and non-line-of-sight (NLOS) error analysis model, which avoids discussing the complex multipath number of paths and its modeling process; in addition, we also propose a multimodal Gaussian noise-based interference model and analyze and model the LEO satellite orbital disturbance. The final simulation results show that our proposed algorithm can not only effectively overcome inertial navigation system (INS) divergence, but also achieve high positioning accuracy, especially when continuous ranging values are used. It can still ensure good anti-interference performance and robustness in terms of path and noise interference and by alternately switching ranging, there are other potential advantages. Compared to some of the existing representative advanced algorithms, it has higher accuracy, stronger stability and lower cost. Furthermore, it can be used as a location reference solution for real-time location services and life search and rescue in harsh environments with incomplete visual satellites and can also be used as a technical reference design solution for the future integration of communication and navigation (ICN).

**Keywords:** 3-satellite; integrated navigation; LEO; INS; switching

## 1. Introduction

Accurate location services in harsh environments are one of the main obstacles to automatic driving, unmanned driving and even search and rescue services. Therefore, high-precision real-time location services in harsh environments have become a research topic of interest in recent years. Reference [1] discussed how to use a low-earth-orbit (LEO) constellation for enhanced navigation and analyzed a series of advantages in the LEO constellation, but did not give a specific design plan for navigation and positioning based on LEO constellations. Reference [2] studied, at the signal level, and proposed an algorithm for navigation based on the differential carrier phase measurement of the LEO satellite signal, but the algorithm did not consider that the time for the LEO satellite to pass the zenith is relatively short and requires frequent switching. Reference [3] used the highly dynamic characteristics of LEO satellites and used the Doppler frequency shift as a measurement value, introducing a Doppler-based positioning algorithm; similarly, the algorithm did not consider the issue of satellite switching. Reference [4], on the basis of a dual-satellite positioning system and a synchronous navigation satellite, a 3-satellite positioning system was proposed. However, it also needed radar altimeter or barometer information to provide complete location services. For 3-satellite positioning technology, References [5–7] provided an algorithm for combining 3-satellite positioning and a strapdown inertial navigation system (SINS), but it also required the help of elevation

information to complete the positioning. Based on radio navigation satellite system/radio determination satellite service (RNSS/RDSS) combined service pseudorange observations, Reference [8] proposed an integrated method that could be connected to a variety of navigation sensors, such as global navigation satellite systems (GNSSs), sensors and INSs, which improved the positioning accuracy in challenging environments, but the use of multiple sensors undoubtedly increased the cost and complexity of the system.

For land vehicle navigation, when various signals block the GPS, especially when the number of visible satellites is not complete, References [9,10] gave a loosely coupled navigation scheme, but this method required an odometer to provide velocity information and multiple sensors were also necessary. Reference [11] combined GPS and vision-based measurements to explore the feasibility of navigation in harsh environments, but the monocular camera used was limited by the weather, so this program could not provide services in inclement weather or at night. Reference [12] gave a regional positioning system solution based on satellite communication to establish a complete system with 3 GEO (Geosynchronous Earth Orbit communication satellite, GEO) + 3 DGEO (Decommissioned Geostationary Orbit communication satellite, DGEO) and 3 IGSO (Geostationary Orbit communication satellite, IGSO). The disadvantage of this solution was that it did not incorporate the three LEO solutions. Reference [13] discussed the feasibility of using a high-precision integrated chip-level atomic clock to assist GPS receivers using three satellites for navigation in urban canyons; however, the price of high-precision atomic clocks was difficult to accept by the general public, thus, limiting the application of the program.

To address the weak GPS environment, for a situation with only three visible satellites, the combination of Doppler measurement and an inertial navigation system (INS) was used to achieve 3-dimensional attitude determination, but the premise was that continuous observation is required [14]. Reference [15] gave an integrated navigation scheme that uses frequency-modulated continuous wave radar (FMCW-Radar) for automatic positioning in harsh environments. Reference [16] proposed a new tightly integrated navigation method consisting of a SINS and a pressure sensor (PS), in which beam measurements are used without converting them to 3D velocity in harsh GNSS environments. Referencing [17] for single satellite navigation and positioning in challenging environments, a navigation and positioning algorithm based on clock bias elimination was given, but no corresponding solutions were given for other positioning situations.

At present, there are advanced algorithms that provide feasible reference solutions for the low-cost and high-precision positioning accuracy of integrated navigation in challenging environments. Reference [18] proposed an application data fusion algorithm for indirect centralized (IC) integrated SINS/GNSS. The main purpose was to improve the positioning accuracy, performance and reliability of low-cost SINS/GNSS integrated navigation systems. The authors of [19–21] proposed a low-cost, high-precision multisensor fusion navigation and positioning design for challenging multipath and non-line-of-sight (NLOS) environments. Reference [22] proposed a new positioning method to improve accuracy in challenging environments by adapting to the random noise of microelectromechanical systems (MEMS-INS) and accurately estimating INS errors. However, this type of algorithm usually needs to observe four satellites.

The newly proposed algorithm does not rely on altimeters and continuous observations, as it is a dynamic positioning algorithm. Switching between LEO satellites can bring the following three advantages:

(1) In military application scenarios, a certain degree of anti-interference ability can be guaranteed by switching. The shorter the switching time is, the less likely it is to be interfered with by the enemy.

(2) Since the bandwidth resource is also an important frequency band resource, switching between LEO satellites can avoid occupying the bandwidth for a long time, thereby making full use of the bandwidth resource and avoiding bandwidth waste.

(3)     Through switching, the LEO satellite navigation service can be used as long as possible while ensuring the LEO communication function. This is mainly based on the ICN perspective.

With the gradual deployment of LEO satellites, such as SpaceX, OneWeb and Hongyan (China), our algorithm has practical significance and practical value. It is a low-cost, low-complexity and anti-jamming algorithm that can be used in harsh environments, such as lush forests, canyons, cities with high-rise buildings and high-latitude areas with few visible satellites, without relying on traditional GNSSs. It provides a navigation plan for outdoor travel, expedition, scientific research and field exploration search and rescue personnel.

## 2. Algorithm Principle

The algorithm can be divided into two categories and a variety of scenarios. Among them, the two categories are INS+LEO 3-satellite alternate switching ranging integrated navigation algorithm and integrated navigation algorithm of INS+2-satellite alternate switching ranging under LEO 3-satellite and the two types of algorithms are divided into three scenarios and four scenarios, respectively. We will describe the relevant principles and processes in detail next. In addition, our algorithm adopts clock error elimination technology based on the ICN [17,23]; therefore, the algorithm we describe next does not consider the clock error between the LEO satellite and the aircraft receiver. Similarly, using a similar system, the clock error between LEO satellites and the clock error between LEO satellites and INS are eliminated by default.

### 2.1. INS+LEO 3-Satellite Alternate Switching Ranging Integrated Navigation Algorithm

2.1.1. Algorithm Principle

The idea behind the algorithm is as follows: According to the satellite selection algorithm of geometric dilution of precision (GDOP) [24], select three satellites that are always visible during the flight of the carrier aircraft. At any time, the aircraft can obtain the real ranging value of one satellite, which changes with the switching of satellites. The remaining two satellites provide virtual ranging values, which also change with the switching of satellites. Then, the ranging values of three LEO satellites cooperate with INS to perform unscented Kalman filter (UKF)-integrated navigation filtering by switching alternately at any time of aircraft flight. The true ranging value is defined as the distance between the current LEO satellite position calculated by LEO satellite ephemeris and the aircraft's true position. In contrast, the virtual ranging value is defined as the distance between the current LEO satellite position calculated by LEO satellite ephemeris and the aircraft position measured by INS.

The 3-satellite alternate switching ranging algorithm can be divided into three scenarios:

Scenario ①: The three satellites participating in the alternation are all on the same orbital plane. This algorithm positioning scene is shown in Figure 1a.

Scenario ②: Two satellites are in the same orbit and the other satellite is in a different orbit (it can be adjacent). This algorithm positioning scene is shown in Figure 1b.

Scenario ③: All three satellites are in different orbits (may be adjacent to each other). This algorithm positioning scene is shown in Figure 1c.

We give a specific algorithm diagram of scenario ①, as shown in Figure 2. Scenario ③ is similar to scenario ①, so it is not given here because of space limitations.

In scenario ①, at any time during the flight of the aircraft, we select three LEO satellites that are visible at all positions and name them PRN1, PRN2 and PRN3. Among these three ranging values, we use the true ranging value of one LEO satellite and the other two LEO satellites use virtual ranging values. Subsequently, the real ranging value and the virtual ranging value are alternately switched with a specific alternate time as the period to realize INS+LEO 3-satellite alternate switching ranging/INS-integrated navigation positioning.

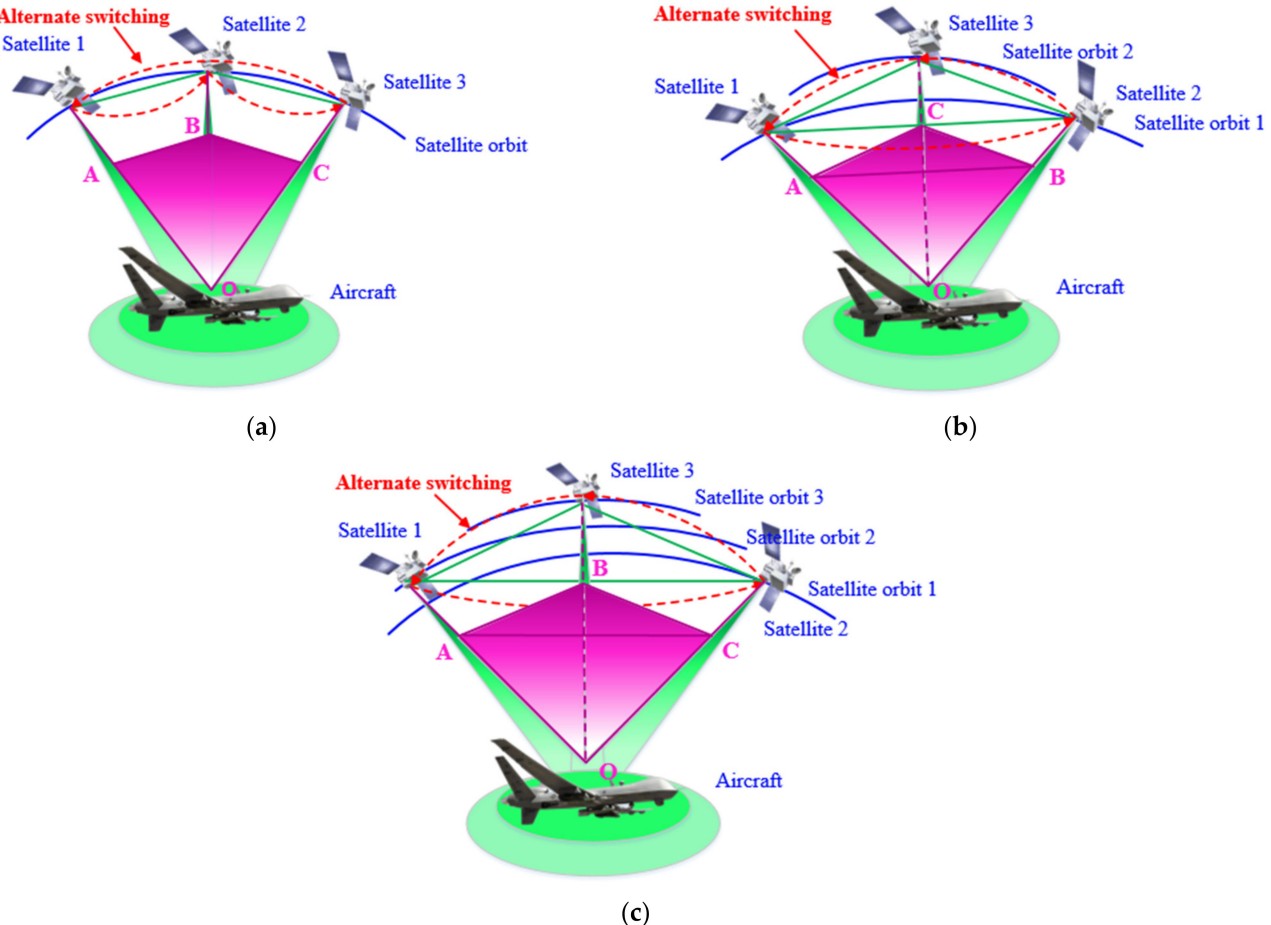

**Figure 1.** Schematic diagram of the INS+LEO 3-satellite. (**a**) Three satellites are in the same orbit. (**b**) Two satellites are in the same orbit. (**c**) Three satellites are in different orbits.

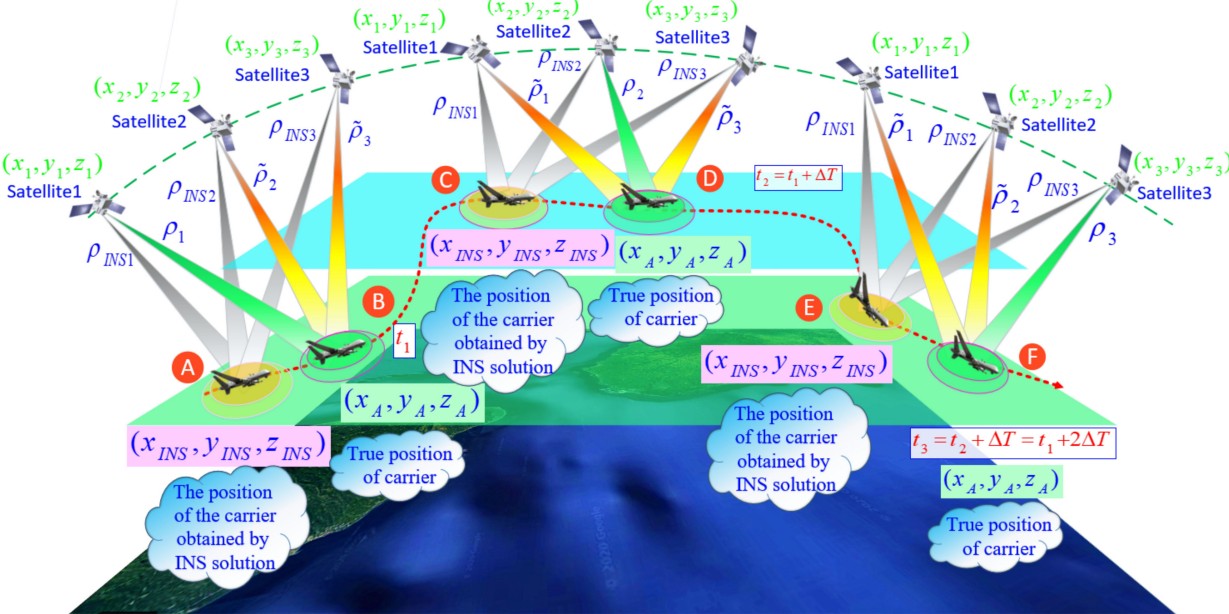

**Figure 2.** 3-satellite alternate switching ranging scene integrated navigation algorithm.

2.1.2. Algorithm Flow

We assume that the position of PRN1 in the Earth-centered Earth-fixed (ECEF) coordinate system is $(x_1, y_1, z_1)$, the position of PRN2 in the ECEF coordinate system is $(x_2, y_2, z_2)$ and the position of PRN3 in the ECEF coordinate system is $(x_3, y_3, z_3)$. The aircraft's real position in the ECEF coordinate system is $(x_A, y_A, z_A)$ and the aircraft's coordinates in the ECEF coordinate system measured by INS are $(\tilde{x}_A, \tilde{y}_A, \tilde{z}_A)$. The position obtained by the INS solution is $(x_{INS}, y_{INS}, z_{INS})$ and the alternate interval is $\Delta T$. $\rho_1, \rho_2$ and $\rho_3$ are the true ranging values of PRN1, PRN2 and PRN3, respectively, and $\tilde{\rho}_1, \tilde{\rho}_2$ and $\tilde{\rho}_3$ signify the virtual values of PRN1, PRN2 and PRN3 measured by INS, respectively. The ranging values $\rho_{INS1}, \rho_{INS2}$ and $\rho_{INS3}$ symbolize the virtual ranging values of PRN1, PRN2 and PRN3 obtained by the INS solution, respectively.

At moment $t_1$, we use the real ranging value $\rho_1$ of PRN1, PRN2 uses the virtual ranging value $\tilde{\rho}_2$ and PRN3 uses the virtual ranging value $\tilde{\rho}_3$. Then, we use the difference $\dot{\rho}_1 = \rho_1 - \rho_{INS1}$ between $\rho_1$ and $\rho_{INS1}$, the difference $\dot{\tilde{\rho}}_2 = \tilde{\rho}_2 - \rho_{INS2}$ between $\tilde{\rho}_2$ and $\rho_{INS2}$ and the difference $\dot{\tilde{\rho}}_3 = \tilde{\rho}_3 - \rho_{INS3}$ between $\tilde{\rho}_3$ and $\rho_{INS3}$, as the UKF filtered observations for filtering. Similarly, at $t_2 = t_1 + \Delta T$ and $t_3 = t_2 + \Delta T = t_1 + 2\Delta T$, we use the real ranging values of $\rho_2$ and $\rho_3$, respectively, and the remaining satellites use virtual values. Finally, we loop this process and switch between the real and virtual ranging value of the satellite at intervals of $\Delta T$ until the flight time of the aircraft ends.

## 2.2. Integrated Navigation Algorithm of INS+2-Satellite Alternate Switching Ranging under LEO 3-Satellite

### 2.2.1. Algorithm Principle

The idea of the algorithm is as follows: according to the satellite selection algorithm of GDOP, three satellites that are always visible during the flight of the carrier aircraft are selected. However, the difference is that, at any time, the aircraft can obtain the true ranging values of two satellites and they will also change with the switching of satellites. The true ranging values of one satellite will not participate in the switching and the true ranging value of another satellite is alternately switched with the virtual ranging value of the remaining satellite. The virtual ranging value also changes with satellite switching, but the three ranging values also participate in INS-integrated navigation filtering.

For 2-satellite alternate switching ranging/INS under the LEO 3-satellite algorithm, the algorithm can be divided into four scenarios:

Scenario ①: The three satellites participating in the alternation are all located on the same orbital plane; one of the satellites uses the continuous true value and the other two satellites alternate ranging with real ranging value and virtual ranging value. This algorithm positioning scenario is shown in Figure 3a.

Scenario ②: Alternately switch satellites in the same orbit and continuous ranging satellites in different orbits (which can be adjacent). This algorithm positioning scenario is shown in Figure 3b.

Scenario ③: The continuous ranging satellite is in the same orbit as an alternate satellite and the other alternate satellite is in a different orbit (it can be adjacent). This algorithm positioning scenario is shown in Figure 3c.

Scenario ④: The three satellites are in different orbits (can be adjacent) and the continuous ranging satellite can be in any orbit. This algorithm positioning scenario is shown in Figure 3d.

We give a specific schematic diagram of scenario ①. As shown in Figure 4, scenario ④ is similar to scenario ①. Due to space limitations, it is not reported in this paper.

### 2.2.2. Algorithm Flow

We maintain PRN1 at the true ranging value and do not participate in the alternation. PRN2 and PRN3 are alternate satellites that switch alternately and the alternate interval is $\Delta T$.

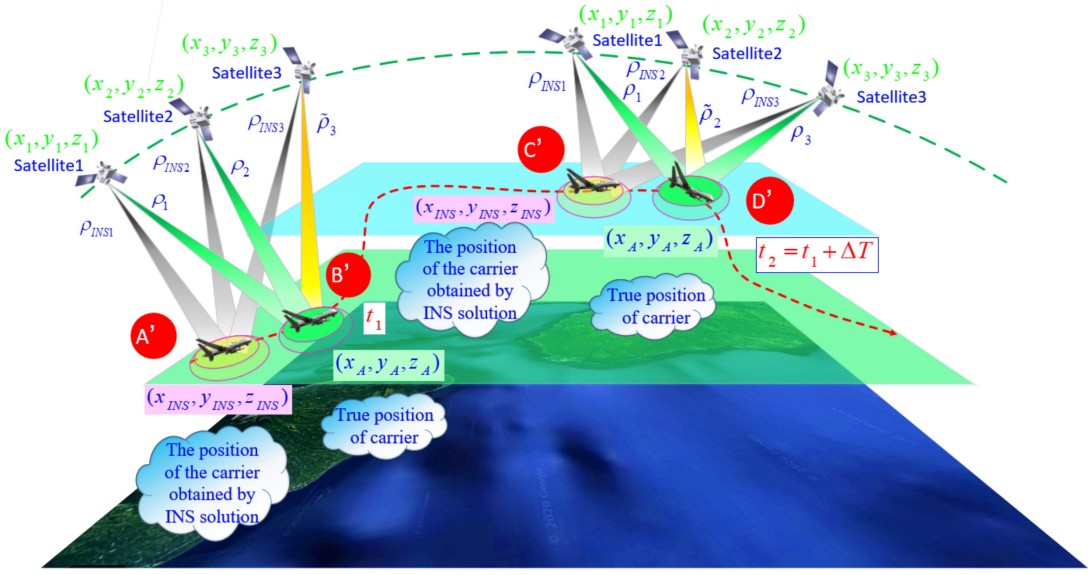

**Figure 3.** Alternate switching ranging scenarios of 2 satellites under 3 satellites under the LEO 3-satellite algorithm. (**a**) Same orbit. (**b**) Alternately switch stars in the same orbit. (**c**) Alternately switch satellites in different orbits. (**d**) Different orbits.

**Figure 4.** Schematic diagram of the integrated navigation algorithm of INS+2-satellite alternate switching ranging under LEO 3-satellite algorithm.

Unlike Section 2.1, at moment $t_1$, we use the real ranging values $\rho_1$ and $\rho_2$ of PRN1 and PRN2 and employ the virtual ranging value $\tilde{\rho}_3$ for PRN3. Then, we use the variation $\dot{\rho}_1 = \rho_1 - \rho_{INS1}$ between $\rho_1$ and $\rho_{INS1}$, the difference $\dot{\rho}_2 = \rho_2 - \rho_{INS2}$ between $\rho_2$ and $\rho_{INS2}$ and the disparity $\dot{\tilde{\rho}}_3 = \tilde{\rho}_3 - \rho_{INS3}$ between $\tilde{\rho}_3$ and $\rho_{INS3}$ as the UKF-filtered observations for filtering.

At moment $t_2 = t_1 + \Delta T$, we use the real ranging values $\rho_1$ and $\rho_3$ of PRN1 and PRN3 and use the virtual ranging value $\tilde{\rho}_2$ for PRN2. The next processing flow is similar to the previous step. Finally, loop this process and alternate switch the LEO satellite's real ranging value and virtual ranging value every time $\Delta T$ elapses until the aircraft's flight time ends.

## 3. Ranging Error Source Analysis and Modeling

This section may be divided by subheadings. It should provide a concise and precise description of the experimental results, their interpretation, as well as the experimental conclusions that can be drawn.

### 3.1. Satellite Ephemeris Error Analysis and Modeling

Since we assume that the clock error has been eliminated, we do not need to consider the clock error parameter. The broadcast ephemeris error three-dimensional root mean square (RMS) model is [25]:

$$\mathrm{RMS}_{Eph} = \sqrt{\sum_{k=1}^{n} \frac{R_k^2 + T_k^2 + N_k^2}{n}} \tag{1}$$

where $R$, $T$ and $N$ represent the orbital component errors in the radial, tangential and normal directions, respectively; the radial component represents the direction of the connection between the satellite and the receiver and the tangential component points to the velocity of the satellite; and the normal component points in a direction perpendicular to the orbital plane of the satellite. The signal-in-space range error (SISRE) calculation equation is often used to measure the accuracy of broadcast ephemeris and clock error parameters. Since the clock bias has been eliminated by default, we only consider the influence of the orbit, then the broadcast ephemeris orbit error and SISRE can be expressed by the following equations [26]:

$$\mathrm{URE}_{orb} = \sqrt{[A_R \mathrm{RMS}(R)]^2 + A_{T,N}^2 [\mathrm{RMS}^2(T) + \mathrm{RMS}^2(N)]} \tag{2}$$

$$\mathrm{SISRE}_{orb} = \sqrt{[\mathrm{RMS}(A_R R)]^2 + A_{T,N}^2 [\mathrm{RMS}^2(T) + \mathrm{RMS}^2(N)]} \tag{3}$$

where $T$ is the trace error, $N$ is the normal error, $R$ is the radial error and $A_R$ and $A_{T,N}^2$ are the contribution factors of the SISRE, which are calculated from the projection of each error in the direction of the average pseudorange. Table 1 shows the contribution values of different types of systems [25,27].

**Table 1.** SISRE contribution factors of different constellation types.

| System Name | Orbit Inclination (deg) | Orbit Height (km) | $A_R$ | $A_{T,N}$ | Attrib-Utes |
|---|---|---|---|---|---|
| Telesat | 99.5 | 1000 | 0.6007 | 0.5653 | LEO |
| Hongyan | null | 1100 | 0.6007 | 0.5653 | LEO |
| SpaceX | 53 | 1150 | 0.6091 | 0.5607 | LEO |
| OneWeb/Telesat | 87.9/37.4 | 1200 | 0.6176 | 0.5562 | LEO |
| GLONASS | 64.8 | 19,100 | 0.9775 | 0.1491 | MEO |
| GPS | 55 | 20,200 | 0.9794 | 0.1427 | MEO |
| BDS-3 | 55 | 21,500 | 0.9810 | 0.1360 | MEO |
| Galileo | 56 | 23,222 | 0.9800 | 0.1210 | MEO |
| BDS-GEO/IGSO | 0/55 | 35,786 | 0.9920 | 0.0880 | GEO/IGSO |

### 3.2. The Error Model of Ionospheric and Tropospheric

Research on the error model of ionospheric and tropospheric propagation is relatively mature. Therefore, we refer to references [28,29] to directly give the ionospheric and convective error correction model as follows:

$$\sigma_{Ionosphere} \sim \left[1 - \left(\frac{R_e \cos \varphi}{R_e + h_{ave}}\right)^2\right]^{-1/2} \tag{4}$$

$$\sigma_{Troposphere} \sim \left[1 - \left(\frac{\cos \varphi}{1.001}\right)^2\right]^{-1/2} \tag{5}$$

where $R_e$ is the average radius of the Earth and $h_{ave}$ is the average height of the ionosphere. $\varphi$ is the elevation angle of the satellite.

### 3.3. Multipath and Non-Line-of-Sight Error Analysis and Modeling

When a satellite signal has a multipath phenomenon, regardless of NLOS reception, the receiving antenna receives the direct signal and several multipath signals, so the radio frequency received signal processed by the receiver is the superposition of these direct signals and multipath signals [30]. However, in the actual environment, especially in harsh and challenging environments, the LOS path of radio propagation is blocked by obstacles, such as mountains, bushes and buildings, and radio waves can only be refracted or reflected; other NLOS dissemination methods can be used to spread the signal.

Generally, the receiver cannot distinguish between direct signal and multipath signal and the receiver loop will directly track and lock the composite signal; therefore, we propose the following compound measurement error model:

$$\varepsilon_{all} = \varepsilon_{\text{LOS}/mult} + \varepsilon_{\text{NLOS}} \tag{6}$$

where $\varepsilon_{\text{LOS}/mult}$ is the error caused by multipaths in the LOS environment and $\varepsilon_{\text{NLOS}}$ is the error caused by the NLOS. In the NLOS environment, its value is a positive number.

We define the ratio of the sum of the error caused by multipath influence in the LOS environment and the error caused by NLOS to the amplitude of the direct signal (MLNSR) to describe the influence of multipath and NLOS ranging errors of the signal, which is a dimensionless relative value. The definition is as follows:

$$\text{MLNSR} = \frac{A_{all}}{A_S} = \frac{A_{\text{LOS}/muti} + A_{\text{NLOS}}}{A_S} \tag{7}$$

where $A_{all}$ is the combined amplitude of the amplitude $A_{\text{LOS}/muti}$ of the error signal caused by multipath effects in the LOS environment and the amplitude $A_{\text{NLOS}}$ of the error signal caused by NLOS and $A_S$ is the amplitude of the direct signal.

In this way, we can analyze the MLNSR without having to discuss the number of complex multipath paths and how to model them. In particular, when MLNSR = 0, we believe that there is only a direct signal and no multipath or NLSO signal interference or that the error signal caused by multipaths in the LOS environment cancels with the error signal caused by NLOS, which is an ideal navigation situation. Generally, MLNSR > 0 and its size are related to external environmental factors.

### 3.4. Analysis and Modeling of Noise Interference

In navigation and signal processing theory, noise models are usually based on Gaussian noise. In general, electronic equipment and interference environments, it is accurate enough to model noise as Gaussian noise. However, in some harsh interference environments or extremely complex environments, much noise that affects navigation reception is non-Gaussian noise. Therefore, combined with the application needs of our algorithm, we

model the noise as multimode Gaussian noise to simulate a more realistic and challenging noise environment. We consider two multimode Gaussian noise models below.

Since an extreme form of narrowband interference is continuous wave (CW) interference in the form of sines and cosines [31], this model can be used to simulate the interference environment of noise plus narrowband interference; the other form is Gaussian-process-superimposed signal interference.

(1) For the Gaussian process plus sine or cosine oscillation process, the probability density function of this model is [32]:

$$p(x) = \frac{1}{(2\pi)^m \sqrt{2\pi}\sigma_0} \int_0^{2\pi} \cdots \int_0^{2\pi} \exp[-\frac{(x - \sum\limits_{i=1}^m B_i \cos\theta_i)^2}{2\sigma_0^2}] d\theta_1 \cdots d\theta_m \tag{8}$$

where $\sigma_0^2$ is the variance of the Gaussian component, $B_i$ is the amplitude of the *i*-th sine or cosine oscillating signal and $\theta_i$ is the phase of the *i*-th signal. We obtain the average power of noise from Equation (8) as:

$$P = \sum_{i=1}^m \frac{B_i^2}{2} + \sigma_0^{2m}, \ i \in N \tag{9}$$

At this time, we define the signal-to-noise ratio (SNR) as the ratio of the average power of the signal to the average power of the noise. The average power of the signal is $P_S$; then:

$$\text{SNR} = \frac{P_S}{P} = \frac{P_S}{\sum\limits_{i=1}^m \frac{B_i^2}{2} + \sigma_0^{2m}}, \ i \in N \tag{10}$$

(2) For the interference between Gaussian-process-superimposed signals, the probability density function of this model is [33]:

$$p(x) = \frac{1}{\sqrt{2\pi}\sigma_0} \sum_{i=0}^m q_i \exp[-\frac{(x - b_i)^2}{2\sigma_0^2}] \tag{11}$$

where $\sum\limits_{i=1}^m q_i = 1$ is the probability of inter-signal interference and its distribution can be a uniform or binomial distribution [32].

For Equation (11), we can find the average power of noise as:

$$P = \sum_{i=1}^m q_i b_i^2 + \sigma_0^2, \ i \in N \tag{12}$$

At this time:

$$\text{SNR} = \frac{P_S}{P} = \frac{P_S}{\sum\limits_{i=1}^m q_i b_i^2 + \sigma_0^2}, \ i \in N \tag{13}$$

Comparing Equations (9) and (12) or (10) and (13), we can find that when $i = 1$ and $B_i = \pm B$, the multimode interference at this time is dual-modal interference and they have general expressions. The corresponding SNR can be expressed by the following equation:

$$\text{SNR} = \frac{P_S}{P} = \frac{P_S}{\beta^2 + \sigma_0^2}, \ \beta = b, B \tag{14}$$

We further sort out:

$$\text{SNR} = \frac{P_S}{P} = \frac{1}{(\frac{\beta^2}{\sigma_0^2} + 1)} \times \frac{P_S}{\sigma_0^2} = \frac{1}{(C_\beta + 1)} \times \text{SNR}_0, \quad \beta = b, B \tag{15}$$

where $C_\beta$ is the ratio of interference noise to Gaussian noise intensity, which reflects the relative strength of interference noise and dual-mode Gaussian noise. It is also the existence of $C_\beta$ that leads to a decrease in the entire SNR and $SNR_0$ is the well-known Gaussian noise, that is, single-mode Gaussian noise. Therefore, the larger $C_\beta$ is, the greater the impact on the SNR of the entire system.

### 3.5. Analysis and Modeling of LEO Satellite Orbit Disturbance

LEO satellites are affected by various forces during their movement around the Earth. These forces can be divided into two categories: conservative forces and nonconservative forces (divergent forces). The former can be described by a "potential function", while the latter force system does not have a "potential function" and can only use the expressions of these forces directly. Applying Newton's second law to obtain the motion equation of an artificial satellite is as follows [34]:

$$\ddot{r} = F_{TB} + F_{NB} + F_{NS} + F_{TD} + F_{RL} + F_{SRP} + F_{AL} + F_{DG} + F_{TH} \tag{16}$$

where $F_{TB}$ is the two-body gravitation, the attraction of the center of the Earth to the LEO satellite; $F_{NB}$ is the gravitational attraction of the sun, moon and other planets, except the Earth on the LEO satellite; $F_{NS}$ is the gravitational attraction of the non-spherical part of the Earth to the LEO satellite; $F_{TD}$ is the change in the Earth's gravitational force on the LEO satellite caused by the Earth's tide; $F_{RL}$ is the influence of the relativistic effect; $F_{SRP}$ is the force of sunlight pressure on the LEO satellite; $F_{AL}$ is the pressure of the Earth's infrared radiation and the Earth's reflected light on the LEO satellite; $F_{DG}$ is the resistance of the Earth's atmosphere to LEO satellites; and $F_{TH}$ is other forces acting on the LEO satellite, such as the satellite attitude control force.

For a specific problem, we do not need to consider all the mechanical factors. When the accuracy requirements (denoted as $\kappa^*$) are given, the corresponding perturbation factors should be estimated:

$$\kappa = \frac{F_\kappa}{F_0} \tag{17}$$

where $F_\kappa$ is a certain kind of perturbation force mentioned above, $F_0$ is the central gravity and $\kappa$ is the magnitude of the perturbation force relative to the gravity of the central body. Table 2 [34] shows the main perturbation magnitude data for four typical GEO satellites, IGSO satellites, MEO satellites and LEO satellites.

**Table 2.** The magnitude of the perturbation force of several typical orbiting satellites.

| | GEO | IGSO | MEO | LEO |
|---|---|---|---|---|
| Earth aspheric perturbation | $3.68 \times 10^{-5}$ | $4.00 \times 10^{-5}$ | $9.27 \times 10^{-5}$ | $1.59 \times 10^{-3}$ |
| Solar gravitational perturbation | $1.05 \times 10^{-6}$ | $7.42 \times 10^{-6}$ | $2.96 \times 10^{-6}$ | $4.34 \times 10^{-8}$ |
| Lunar gravitational perturbation | $2.02 \times 10^{-5}$ | $2.10 \times 10^{-5}$ | $5.73 \times 10^{-6}$ | $7.48 \times 10^{-8}$ |
| Solar pressure perturbation | $4.49 \times 10^{-8}$ | $4.49 \times 10^{-8}$ | $1.93 \times 10^{-8}$ | $1.01 \times 10^{-9}$ |
| Solid tidal perturbation | $3.19 \times 10^{-10}$ | $1.69 \times 10^{-10}$ | $4.01 \times 10^{-10}$ | $7.34 \times 10^{-9}$ |
| Perturbation of relativistic effect | $3.17 \times 10^{-10}$ | $3.17 \times 10^{-10}$ | $4.82 \times 10^{-10}$ | $2.01 \times 10^{-9}$ |
| Atmospheric drag perturbation | $5.04 \times 10^{-21}$ | $5.42 \times 10^{-15}$ | $7.20 \times 10^{-15}$ | $3.02 \times 10^{-7}$ |

We follow the principle of selection of perturbation factors given in references [35]. For the SpaceX constellation, we select the perturbation accuracy $\kappa^* = 10^{-7}$. For conservative and nonconservative forces, there are the following selection principles:

① If the perturbation force is conservative and if it satisfies

$$2\kappa(n\Delta t) \geq \kappa* \tag{18}$$

then the perturbation factor must be considered. For the situation that only causes a short period of change, the condition becomes:

$$2\kappa \geq \kappa* \tag{19}$$

② If the perturbation force is a nonconservative force (dissipation force) and if it satisfies

$$\frac{3}{2}\kappa(n\Delta t)^2 \geq \kappa* \tag{20}$$

where $n\Delta t$ is the arc length experienced by the satellite movement.

According to Table 2 and combined with the above selection rules on perturbation factors, we focus on the Earth's non-spherical perturbation and atmospheric resistance perturbation. Mathematical models of the Earth's nonspherical perturbation and atmospheric drag perturbation are very complicated. In addition, in view of the satellite orbit calculation, the accuracy of the perturbation calculation is not high and the speed of the perturbation calculation is required to be faster. In view of the various uncertainties of the drag effect itself, it is necessary to rely on measured data and statistical analysis to reduce it. However, SpaceX has not yet completed global deployment and relevant information is not publicly available; therefore, it is difficult for us to obtain public information about SpaceX to establish the Earth's nonspherical perturbation. Therefore, we refer to the research results of LEO satellite orbit perturbation in reference [36] and analyze the following simplified models of the Earth's nonspherical perturbation and atmospheric resistance perturbation:

① Earth nonspherical perturbation model

The change in the position offset caused by the nonspherical gravitational perturbation of the Earth is periodic and its period is close to the operating period of the LEO satellite. However, the main trend of the offset is an approximately linear increase and the LEO satellites increase by approximately 32 km for each revolution. We give the following mathematical model after parameter coupling:

$$F_{NS} = \{\alpha[\sin(\frac{2\pi}{T_{\text{LEO}}}t + \theta)(1 + kt)] + N\alpha_0\} * (1 + \kappa * \sigma_{rand}) \tag{21}$$

where the value of $\alpha$ is based on the offset of the satellite position after 24 h of the Earth's nonspherical gravity in the reference [36] and the value is 478.1518787694562 km; $\theta$ represents the initial phase of the perturbation; here, we only consider the case of $\theta = 0$; $T_{\text{LEO}} = 108$ min, is SpaceX's satellite operation period; $\alpha_0 = 32$ km is the orbit increment of the LEO satellite per revolution; $N$ is the number of revolutions; here, we take $N = 14$; and the term $(1 + kt)$ is mainly the linear increment in the simulated offset. According to reference [36], we can find that $k$ is approximately equal to 0.3320 km/min. In addition, regarding some of the orbital interference caused by internal or external interference, we might also use random noise $\sigma_{rand}$ with a mean value of 0 and a variance of 1 to simulate and use the selected perturbation accuracy $\kappa*$ to describe its amplitude.

② Atmospheric drag perturbation model

The increase in position offset caused by atmospheric drag perturbation is nonlinear and the growth rate will gradually increase over time, which gradually strengthens the influence of atmospheric drag perturbation. Therefore, we fit 160 sets of simulation data extracted from reference [36] and the fitting results are as follows:

$$F_{\text{LEO\_drag}} = 0.003612 + 1.048 \times 10^{-5}t + 2.388 \times 10^{-6}t^2 \tag{22}$$

Then, the mathematical model of the influence caused by the atmospheric drag perturbation after parameter coupling is:

$$F_{DG} = F_{\text{LEO\_drag}} * (1 + \kappa * \sigma_{rand}) \tag{23}$$

Finally, we bring Equations (21) and (23) into Equation (16) and the total perturbation due to the aspherical perturbation of the Earth and the perturbation of atmospheric drag is:

$$
\begin{aligned}
\ddot{r}_0 &= F_{NS} + F_{DG} \\
&= \{[\alpha \sin(\tfrac{2\pi}{T_{\text{LEO}}}t + \theta)(1 + kt) + N\alpha_0] + F_{\text{LEO\_drag}}\} * (1 + \kappa * \sigma_{rand}).
\end{aligned}
\tag{24}
$$

### 3.6. Other Models and Errors

We use the ENU coordinate system as the navigation coordinate system. We use the navigation geographic coordinate system as the ENU coordinate system, taking into account the aircraft's flying height. Simultaneously, the Earth is considered to be a rotating ellipsoid. The error model of the gyroscope and LEO positioning receiver error model can be found in references [17,37,38]. Due to space limitations, we will not elaborate the model here.

Other errors, such as tracking error, mainly involve code tracking error and carrier tracking errors, which are related to specific factors, such as the modulation code type, loop model, phase detector and loop bandwidth used by the system [39,40]. However, the specific navigation design scheme based on the world's major LEO navigation enhancement systems has not been introduced, so we will learn from the relevant parameters of the existing navigation system for setting. In addition, regarding group wave delay and inter-channel differences, in cases, such as pure GLONASS and GNSS joint positioning, using multiple GNSS signals broadcast on different carrier frequencies, it is necessary to consider such measurement errors [41]. Other interference models can be found in references [42,43]. Due to space limitations, we will not introduce them in detail here.

## 4. Simulation Analysis

### 4.1. Simulation Parameters

At present, the representative large-scale global LEO constellations are SpaceX, OneWeb and Telesat. There are also several characteristic LEO constellations under construction, such as Hongyan, Hongyun and Hiber. Without loss of generality, we adopt the SpaceX constellation for simulation analysis. The main parameters of the SpaceX core constellation are shown in Table 3.

**Table 3.** SpaceX core constellation parameters [44].

| Parameter Type | Value |
|---|---|
| Orbit radius (km) | 7521 |
| Height (km) | 1150 |
| Orbit surface number | 32 |
| Inclination (°) | 53 |
| Number of satellites per orbit | 50 |
| Total number of satellites | 1600 |
| Signal ionospheric delay (m) | 0.02 |
| Signal tropospheric delay(m) | 0.001 |
| Signal spatial error (m) | 0.01 |
| Pseudorange measurement noise (m) | 0.75 |
| Pseudorange rate measurement noise(m/s) | 0.045 |
| Code tracking error (m) | 0.1 |
| Range rate tracking error (m/s) | 0.001 |

Table 4 lists the relevant parameters in the inertial measurement unit (IMU) model. Since the UKF algorithm is an excellent nonlinear algorithm, it has a wide range of applications in the navigation field; therefore, our filtering algorithm adopts the UKF algorithm based on UT transformation [45,46].

**Table 4.** IMU main parameters [47].

| Parameter Type | | Accelerometer | Gyro |
|---|---|---|---|
| Quantization level | | 20 (micro-g) | 41 (deg/hour) |
| Noise root PSD | | 2 (micro-g/$\sqrt{Hz}$) | 0.002 ((deg/hour)0.5) |
| Scale Factor | | 100~1000 ($\times 10^{-6}$ ppm) | 100~1000 ($\times 10^{-6}$ ppm) |
| Cross-coupling error | | 100~1000 ($\times 10^{-6}$ ppm) | 100~1000 ($\times 10^{-6}$ ppm) |
| Bias | | [30 45 26] (micro-g) | [0.09 0.013 0.08 ] (deg/hour) |
| Bias uncertainty per instrument | | 30 (micro-g) | 0.001 (deg/hour) |
| Initial uncertainty per axis | Attitude | 0.01 (deg) | 0.01 (deg) |
| | Velocity | 0.01 (m/s) | 0.01 (m/s) |
| | Position | 1 (m) | 1 (m) |

Table 5 shows the initial values of the UKF and aircraft parameters. In addition, the aircraft completes two 45° turns in opposite directions and climbs 500 m simultaneously during the flight; the simulation time is 418 s.

**Table 5.** Main parameters of UKF and aircraft.

| UKF Parameter Type | Value |
|---|---|
| Initialization position error (m) | 0 |
| Initialization velocity error (m/s) | 0 |
| Initialized attitude error (°) | 0 |
| The sampling interval is (s) | 0.01 |
| Initial position | (50.425° N, −3.5958° E) |
| Initialising velocity (m/s) | 200m/s |
| Initial attitude (°) | 0- Roll, 0- Pitch, 90- Yaw |
| Initial altitude (m) | 10,000 |

*4.2. Scenario Description*

The object of our experimental research is an aerial aircraft. We adopt the SpaceX LEO satellite system combined with an INS and the integrated navigation and positioning of the aircraft are realized by alternately switching between three satellites. According to the description in Section 2, we divide the research scenarios into different satellite orbital planes and phases:

(1) INS+LEO3-satellite alternate switching ranging integrated navigation and positioning on the same orbital plane.
(2) INS+LEO3-satellite alternate switching ranging integrated navigation and positioning on different orbital planes;
(3) INS+2-satellite alternate switch ranging under LEO3- satellite on the same orbit integrated navigation and positioning;
(4) INS+2-satellite alternate switch ranging under LEO3- satellite on different orbital integrated navigation and positioning.

At the same time, in scenarios (3) and (4), we divide the research scenarios into original scenarios and comparison scenarios:

(1) The original scenario is INS+ 2-satellite alternate switching ranging under LEO3- satellite integrated navigation and positioning; among them, there is a continuous ranging satellite;
(2) The comparison scenario removes the satellite that is continuously ranging and uses the INS+LEO2- satellite to alternate switching ranging integrated navigation and positioning.

Furthermore, the satellite elevation angle was set to 10°. In the above scenarios, we set the alternate switching time to 5 s, 10 s, 30 s and 60 s according to the different alternate interval times for navigation and positioning research and, finally, the positioning effect is

analyzed. We do not consider the influence of multipath, NLOS and LOS interference for the time being and the noise is modeled as single-modal Gaussian noise.

### 4.3. Experimental Results

4.3.1. INS+LEO 3-Satellite Alternate Switching Ranging Integrated Navigation Algorithm

(1) Based on same orbits

According to the GDOP value, we select the three LEO satellites that are always visible during the aircraft's movement. These three satellites must be in the same orbit. For this experiment, we select three satellites with PRN (pseudorandom noise code) numbers 209, 221 and 245 on the fifth orbit of the SpaceX LEO satellite system and named them PRN1, PRN2 and PRN3, respectively. Finally, the simulation is carried out and the result is shown in Figure 5 (EPE, NPE and UPE represent the position errors of the east, north and up directions, respectively). The meaning of the other parameters, such as EVE and AEE, can be deduced by analogy, so we do not discuss them later.

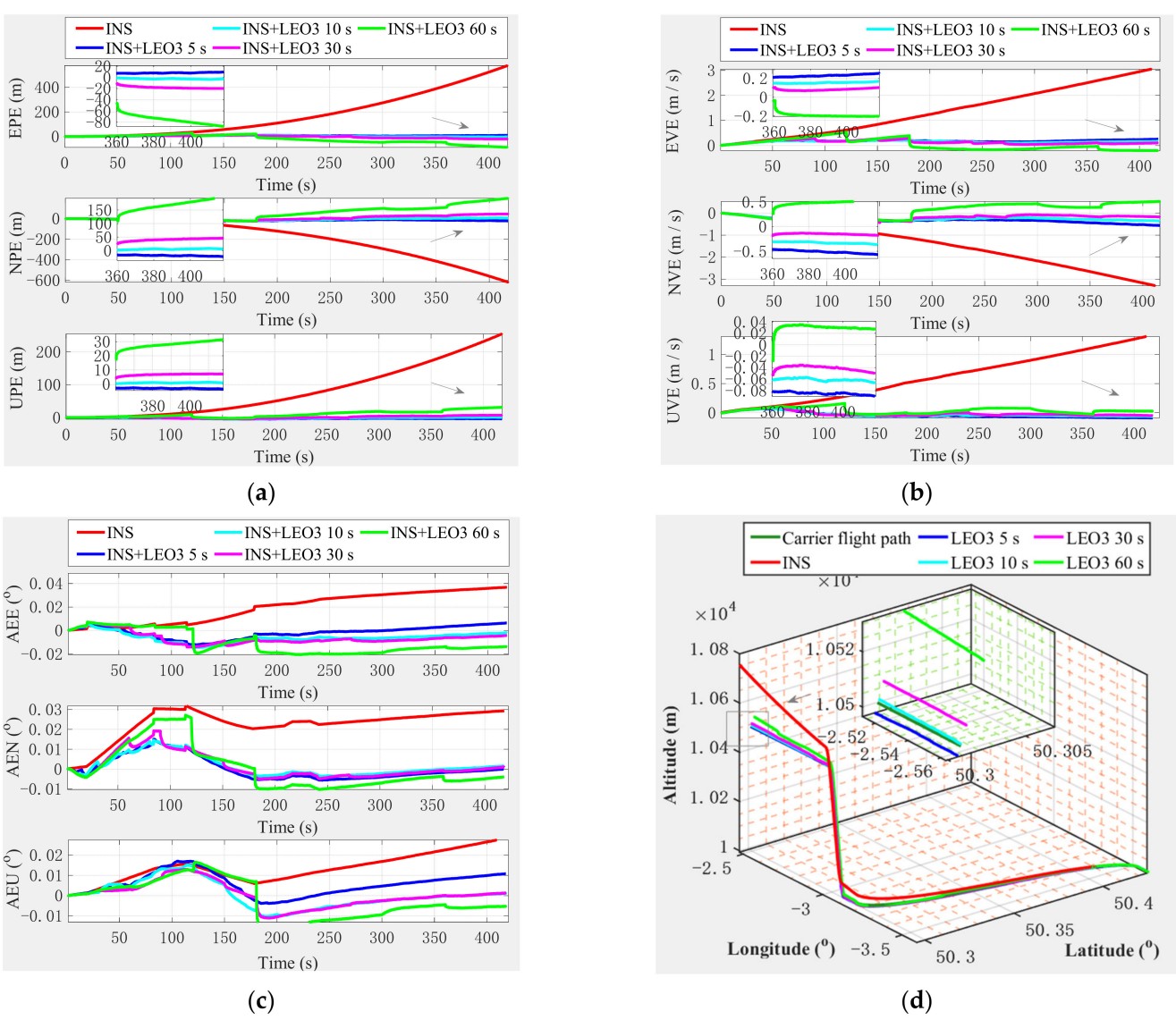

**Figure 5.** Positioning error curve of INS+LEO3-satellite alternate switching ranging integrated navigation based on the same orbit. (**a**) Position error. (**b**) Velocity error. (**c**) Attitude error. (**d**) 3D trajectory error.

In Figure 5, the INS+LEO3-satellite alternate switching ranging integrated navigation algorithm is based on the same orbit. Whether it is a position (Figure 5a), velocity (Figure 5b) or attitude error (Figure 5c), the error can be well restrained compared with INS navigation. The stable values of errors almost all tend to zero. In Figure 5d, the final trajectory curve closely follows the real movement trajectory of the aircraft. As a whole, as the switching time increases, the error increases accordingly. From the effect point of view, we sort the effects in descending order as follows: 5 s > 10 s > 30 s > 60 s; that is, the switching time is the best for 5 s, 10 s and 30 s, but 60 s has the worst effect. However, there are certain fluctuations in the error of the individual switching time, since the IMU is constantly correcting navigation errors.

(2) Based on different orbits

The principle of selecting satellites is the same as in the same orbit. We select three satellites in different orbits that are always visible on the trajectory of the aircraft. In this experiment, three satellites with the fifth orbit plane PRN number 209, the sixth orbit plane PRN number 299 and the eighth orbit plane PRN number 376 satellites were selected for the SpaceX satellite system and they were named PRN1, PRN2 and PRN3. Finally, the simulation is performed and the result is shown in Figure 6.

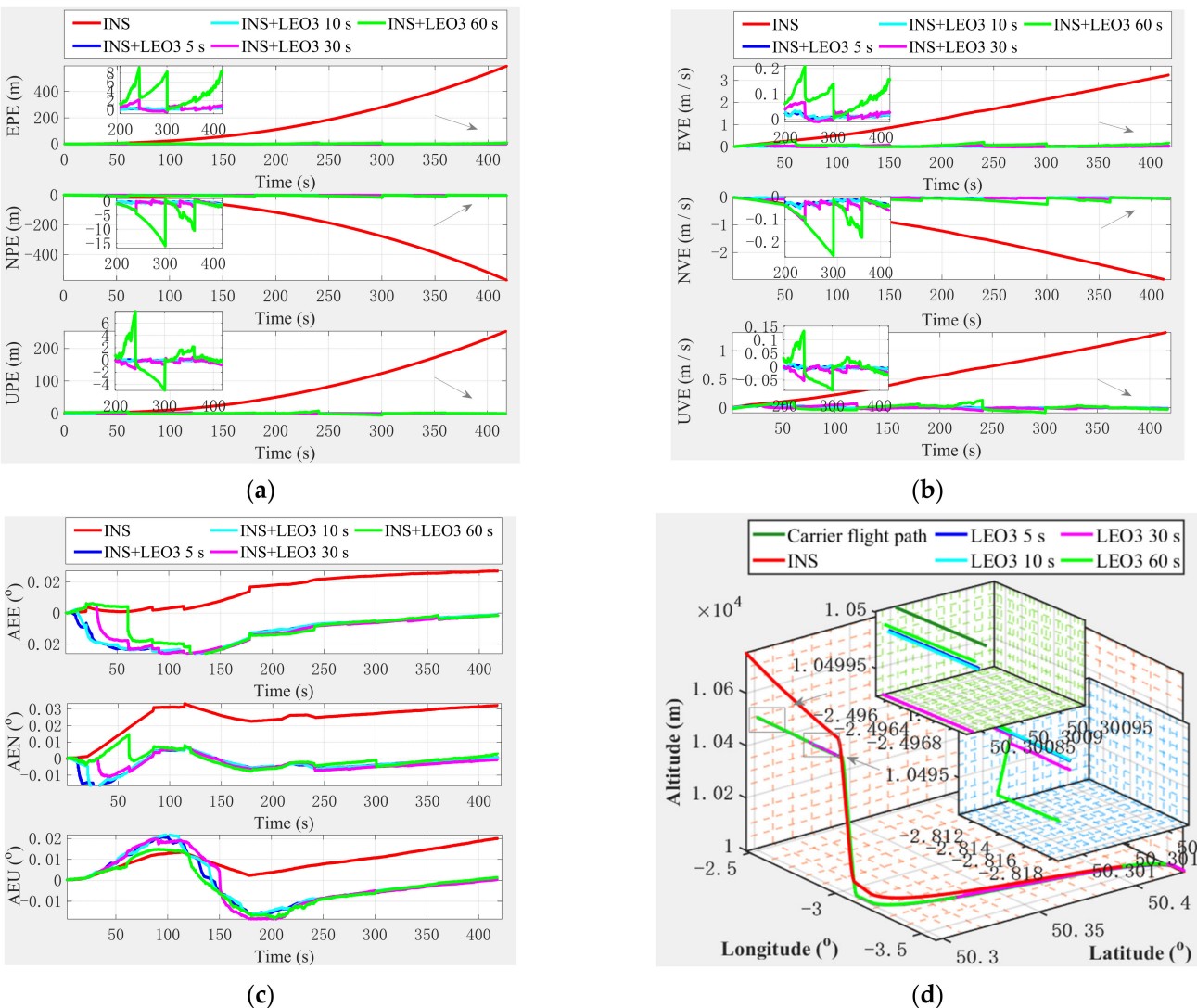

**Figure 6.** Positioning error curve of INS+LEO3 satellite alternate switching ranging integrated navigation based on different orbits, (**a**) Position error. (**b**) Velocity error. (**c**) Attitude error. (**d**) 3D trajectory error.

In Figure 6, the INS+LEO3 satellite alternate switching ranging integrated navigation algorithm based on different orbits can also suppress the divergence of position (Figure 6a), velocity (Figure 6b) and attitude error (Figure 6c) in the INS. Furthermore, in Figure 6d, the final trajectory curve is close to the aircraft's real movement trajectory to a greater extent. From an intuitive perspective, the algorithm based on different orbits is obviously better than the algorithm based on the same orbit compared to Figure 5a–d and the error fluctuation is much smaller. This can be observed from each error and the final trajectory error curve. As the switching time increases, the error gradually increases; among them, the error with a switching time of 60 s fluctuates relatively greatly. Nevertheless, it can overcome the divergence in the INS.

### 4.3.2. Integrated Navigation Algorithm of INS+2- Satellite Alternate Switching Ranging under LEO 3- Satellite

(1) Based on same orbits

The principle of satellite selection is the same as above. In this experiment, the original scenarios selected are the three satellites with PRN numbers 209, 221 and 245 on the fifth orbital plane of the SpaceX satellite system and they are named PRN1, PRN2 and PRN3, respectively. In the original scenarios, we keep the satellite with a PRN of 209 as a continuous real satellite ranging value and let the two satellites with PRNs of 221 and 245 use virtual ranging values for alternate ranging. By comparing scenario selection, we remove the satellite with PRN number 209, the two satellites with PRN numbers 221 and 245 are named the same as the original scenario, then let the satellites with PRN numbers 221 and 245 alternately switch between the real range value and the virtual range value. The simulation results are shown in Figure 7 (where the OE represents the original scenario experiment, CE represents the comparison scenario experiment, SO represents the same orbit and DO represents different orbits. $\tau$ s-OE/SO and $\tau$ s-CE/SO represent the original scenario and the comparison scenario of the alternate switching experiment of the 2-satellite alternate switching ranging under LEO 3-satellite on the same orbit, respectively, and $\tau$ represents the switching time. $\tau$ s-OE/DO and $\tau$ s -CE/DO have the same meaning as $\tau$ s-OE/SO and $\tau$ s-CE/SO, so we do not discuss them later.).

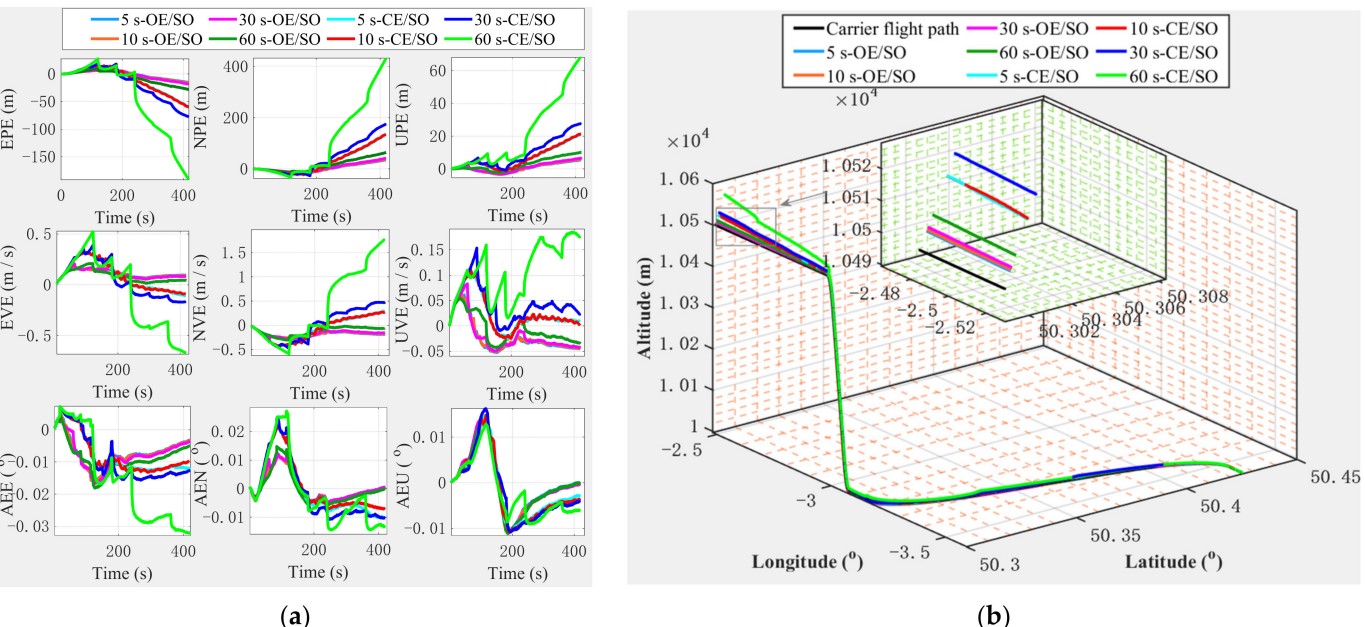

(a)　　　　　　　　　　　　　　　　　　　　　　　　　　(b)

**Figure 7.** Positioning error curve of integrated navigation algorithm of INS+2-satellite alternate switching ranging under LEO 3-satellite on the same orbit. (**a**) Position, velocity and attitude errors. (**b**) 3D trajectory error.

In Figure 7a,b, we can see that under the same orbit, the original scenario and the comparison scenario can also overcome the problem of INS navigation error divergence to a certain extent. However, in the comparison scenario, the effect is significantly worse due to the satellite's removal for continuous ranging. The effect in the original scenario is intuitively similar to that of the 3-satellite alternate switching ranging algorithm under the same orbit, but it is better than the positioning effect in the comparison scenario. Similarly, as the switching time increases, the error gradually increases.

(2) Based on different orbits

The principle of selecting satellites is the same as above. We select three satellites in different orbits that are always visible on the trajectory of the aircraft. In this experiment, the original scenario selected three satellites in the SpaceX satellite system with the fifth orbit PRN number 209, the sixth orbit PRN number 299 and the eighth orbit PRN number 376 and we named them PRN1, PRN2 and PRN3, respectively. In the original scenarios, we keep the satellite with a PRN of 209 as a continuous real satellite ranging value and let the two satellites with the PRN of 299 and 376 use virtual ranging values for alternate ranging. In comparing scenario selection, we remove the satellite with PRN number 209 and let the two satellites with PRNs numbers 299 and 376 alternately switch between the real range value and the virtual range value. The simulation result is shown in Figure 8.

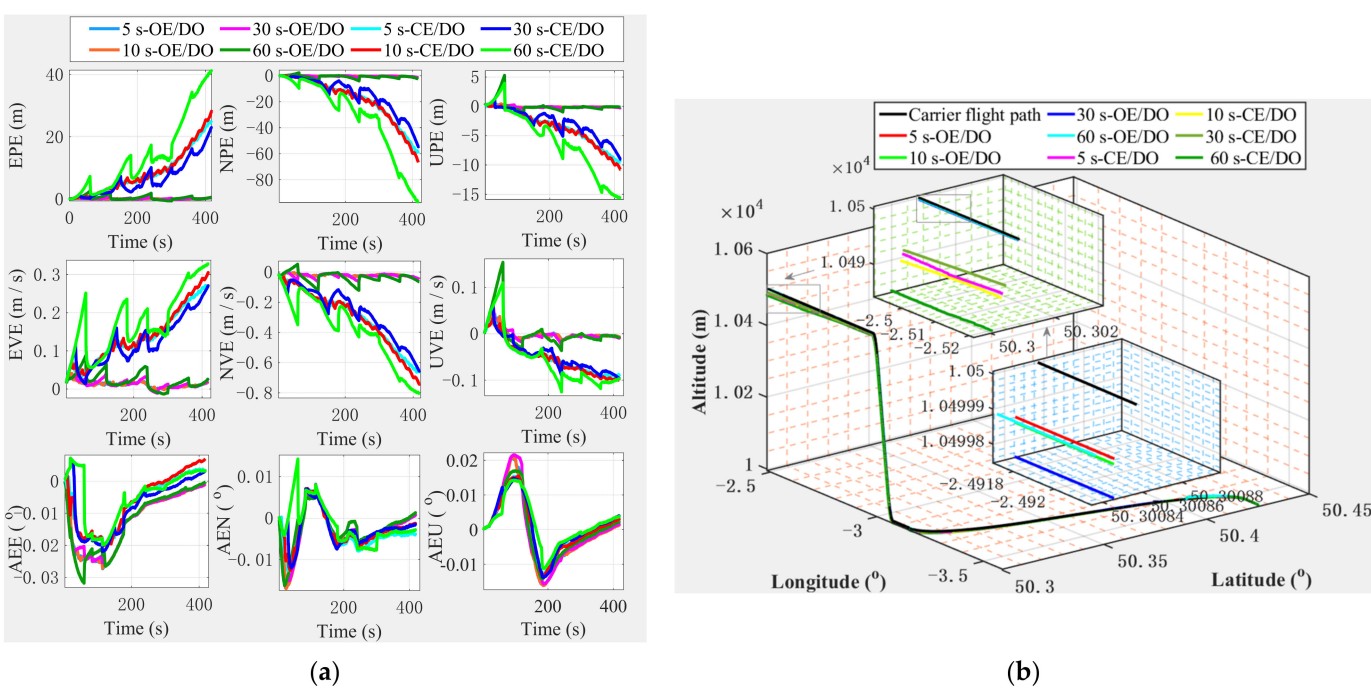

(**a**)　　　　　　　　　　　　　　　　　　　　(**b**)

**Figure 8.** Positioning error curve of integrated navigation algorithm of INS+2-satellite alternate switching ranging under LEO 3-satellite on a different orbit. (**a**) Position, velocity and attitude errors. (**b**) 3D trajectory error.

In Figure 8a,b, we can see that under different orbits, the original scenario and the comparison scenario can overcome INS navigation error divergence to a large extent. Similarly, since the satellite that is continuously ranging is removed from the comparison scenario, the effect is significantly worse. The effect in the original scenario is intuitively similar to the 3-satellite alternate switching ranging algorithm under different orbits and the effect is better than the positioning effect of the comparison scenario under different orbits. Similarly, as the switching time increases, the error gradually increases.

From the above qualitative analysis, we can initially find that algorithms based on different orbits are significantly better than those based on the same orbit, verifying the correctness of judging the pros and cons of positioning accuracy based on the spatial distribution characteristics of satellites proposed in Section 2. Additionally, the original scenario

effect is significantly better than the comparison scenario; that is, we use a continuous real ranging value combined with an alternating real ranging value, which is better than the simple alternating scheme of real ranging value and virtual ranging value. The optimal switching time varies with the system used. Generally, a shorter switching time may be better, but frequent switching increases the power consumption in the receiver and if the switching time is longer, this situation is conducive to satellite communications; however, although the LEO system assists in correcting the INS error, the cumulative effect of the INS error worsens the navigation and positioning effect. Therefore, in actual projects, we have to compromise based on actual business (according to time task) and precision requirements.

### 4.3.3. Comparison of Algorithm Effects in Different Scenarios

We analyze them from a quantitative perspective below to compare the algorithm's effect in different scenarios. Without losing generality, we select the case where the switching time is 5 s as an example for comparative analysis of the algorithm in each scenario. The simulation results are displayed in Figure 9. We also count the corresponding error results listed in Tables 6 and 7. M-Imp signifies the maximum improvement and m-imp denotes the minimum improvement, both of which are relative to INS navigation. We use slanted green fonts and red bold fonts to indicate each indicator's minimum and maximum values.

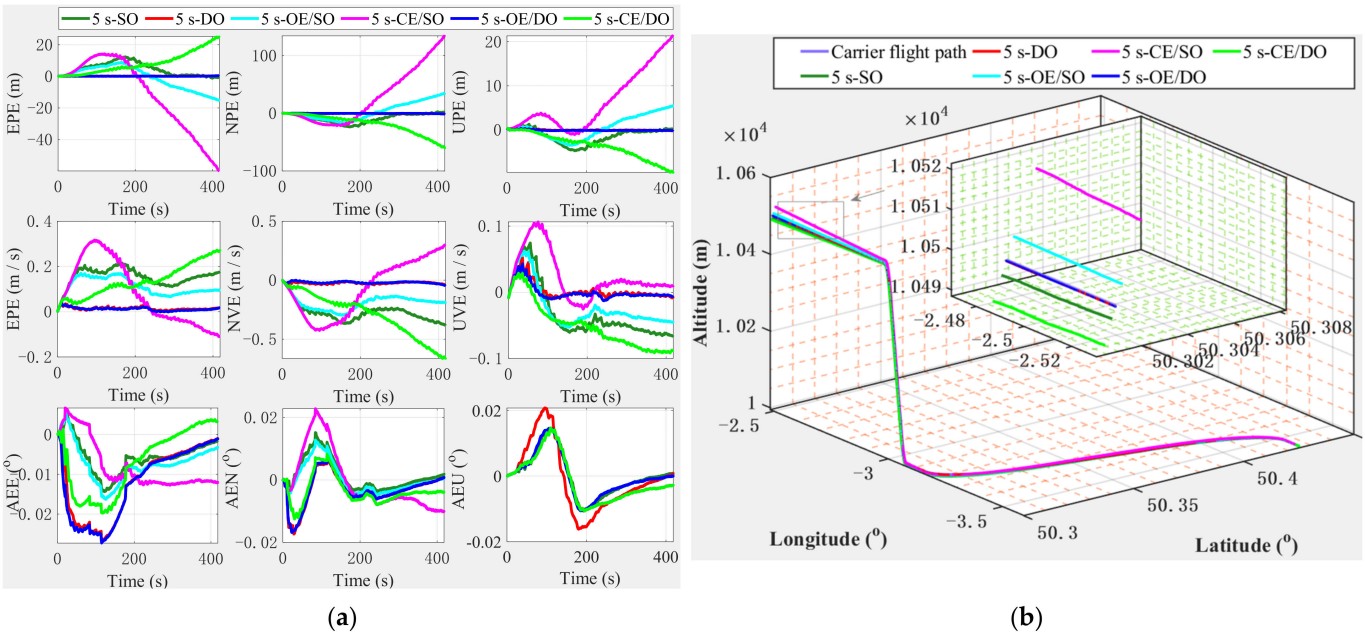

(**a**)　　　　　　　　　　　　　　　　　　　　(**b**)

**Figure 9.** Comparison curve of the algorithm effect of different scenes with a switching time of 5 s. (**a**) Position, velocity and attitude errors. (**b**) Trajectory error.

In Figure 9a,b and Tables 6 and 7, we can see the following:

(1)　Under the same orbits, various algorithms can effectively overcome the divergence in the INS. The performance of these algorithms is ranked from high to low roughly as follows, OE/SO > SO > CE/SO; that is, the performance of the integrated navigation algorithm based on INS+2-satellite alternate switching ranging under the LEO 3-satellite is better than the INS+LEO3-satellite alternate switching ranging algorithm and the original scenario effect is superior to the comparison scenario, which can be seen from the error curves and indicators in Figure 9 and Tables 6 and 7. This result is not difficult to imagine because in the original scenario, we use a continuous real ranging value and let the remaining two satellites alternate ranging, while in the comparison scenario, we remove the real ranging value. For the INS+LEO3-satellite alternate switching ranging algorithm, we use a real ranging satellite to participate in alternate ranging. Therefore, this result is expected.

(2) Under adjacent orbits, various algorithms can also effectively overcome the divergence in the INS. The performance of these algorithms is sorted in descending order as follows: OE/DO > DO > CE/DO; that is, the performance of the integrated navigation algorithm based on INS+2-satellite alternate switching ranging under the LEO 3-satellite is better than that of the INS+LEO3-satellite alternate switching ranging algorithm. The original scenario effect is also apparently better than the comparison scenario, as we can see from the various error curves and indicators in Figure 9 and Tables 6 and 7.

(3) In the same orbits and different orbits and two comparison scenarios, the performance of these algorithms in descending order is roughly as follows: OE/DO > DO > OE/SO > SO > CE/DO > CE/SO.

**Table 6.** The position, speed and attitude error statistics of different scenario algorithms with a switching time of 5 s.

| Error Index | | EPE (m) | NPE (m) | UPE (m) | EVE (m/s) | NVE (m/s) | UVE (m/s) | AEE × 10⁻³ (°) | AEN × 10⁻³ (°) | AEU × 10⁻³ (°) |
|---|---|---|---|---|---|---|---|---|---|---|
| Mean | INS | 182.7123 | −180.3463 | 79.4600 | 1.4360 | −1.3730 | 0.6065 | 14.9527 | 24.6303 | 9.4417 |
| | SO | 4.0566 | −8.2301 | −0.0149 | 0.1539 | −0.2744 | −0.0314 | 5.4351 | 1.4770 | 0.1273 |
| | DO | 0.1237 | *0.0207* | 0.1420 | 0.0124 | −0.0200 | 0.0012 | −1.2575 | −3.3856 | −1.0624 |
| | OE/SO | −0.8131 | 2.8618 | 0.5787 | 0.1113 | −0.1910 | −0.0228 | −7.1301 | 0.2507 | 0.0576 |
| | CE/SO | **−10.9104** | 20.8000 | 6.4129 | **0.1823** | **−0.2212** | 0.0942 | **−7.8222** | −3.2386 | **1.4097** |
| | OE/DO | *0.1152* | −0.2475 | *−0.0681* | 0.0116 | *−0.0198* | *0.0005* | −1.2432 | *−0.0557* | *−0.0446* |
| | CE/DO | 8.2392 | −18.2905 | −3.6055 | 0.1346 | −0.1009 | −0.0507 | 6.7509 | **−3.5034** | 0.9335 |
| M-Imp (%) | | 99.94 | 99.99 | 99.91 | 99.19 | 99.56 | 99.92 | 91.69 | 99.77 | 99.39 |
| m-imp (%) | | 94.03 | 88.47 | 91.93 | 87.31 | 83.89 | 84.47 | 47.69 | 85.78 | 99.53 |
| Std | INS | 175.9062 | 169.1312 | 75.1913 | 0.9700 | 0.8856 | 0.3928 | 10.0270 | 8.1525 | 7.6474 |
| | SO | 4.0433 | 7.7242 | 1.7057 | 0.0384 | 0.0755 | **0.0407** | 4.2212 | 4.9260 | 6.4462 |
| | DO | 0.2142 | 0.5918 | 0.3804 | 0.0099 | 0.0147 | 0.0125 | *0.9125* | 4.9150 | 5.3598 |
| | OE/SO | 7.1943 | 15.0023 | 2.5943 | 0.0375 | 0.0598 | 0.0312 | 4.6051 | 4.6877 | 6.5175 |
| | CE/SO | **22.5312** | 46.6283 | 6.5728 | **0.1383** | **0.2330** | 0.0351 | 5.2373 | **5.4334** | **6.9578** |
| | OE/DO | *0.0674* | *0.2179* | *0.1110* | *0.0069* | *0.0075* | *0.0111* | 4.8759 | *4.1812* | *4.5167* |
| | CE/DO | 6.8376 | 16.1813 | 2.7598 | 0.0686 | 0.1814 | 0.0342 | 8.0577 | 5.5228 | 7.4802 |
| M-Imp (%) | | 99.96 | 99.98 | 99.85 | 99.29 | 99.15 | 97.17 | 90.90 | 48.71 | 25.31 |
| m-imp (%) | | 87.19 | 72.13 | 91.26 | 85.74 | 73.69 | 89.64 | 47.47 | 33.35 | 9.02 |

**Table 7.** Trajectory error statistics of different scene algorithms when the switching time is 5 s.

| Algorithm | Mean | | | Std | | |
|---|---|---|---|---|---|---|
| | Lon × 10⁻⁵ (°) | Lat × 10⁻⁵ (°) | Alt (m) | Lon × 10⁻⁵ (°) | Lat × 10⁻⁵ (°) | Alt (m) |
| INS | 161.8661 | 256.3933 | 79.4600 | 151.7977 | 246.4802 | 75.1913 |
| SO | −7.3871 | 6.3186 | 1.3019 | 6.9331 | 5.6673 | 1.7057 |
| DO | −0.0185 | 0.01734 | −0.0149 | 0.5312 | 0.3002 | 0.3804 |
| OE/SO | 2.5680 | −1.1362 | 0.5787 | 13.4651 | 10.0385 | 2.5943 |
| CE/SO | 24.0537 | −15.2826 | 6.4129 | 41.8501 | 31.5773 | 6.5728 |
| OE/DO | −0.2221 | 1.6158 | −0.0681 | 0.1956 | 0.0944 | 0.1110 |
| CE/DO | −16.4163 | 11.5465 | −3.6055 | 14.5229 | 9.5803 | 2.7598 |
| M-Imp (%) | 99.99 | 99.99 | 99.98 | 99.87 | 99.96 | 99.85 |
| m-Imp (%) | 85.14 | 94.04 | 91.93 | 72.43 | 87.19 | 91.26 |

In general, the 3-satellite alternate switching ranging algorithm based on different orbits and the original scenarios based on the integrated navigation algorithm of INS+2-satellite alternate switching ranging under LEO 3-satellite on different orbits has the best results. It can improve by more than 90% relative to INS in most performance indicators. From the final trajectory curve and parameters, the errors of longitude, latitude and altitude are all very small, which meet the needs of cm level real-time location services. However, the INS + LEO3-satellite alternate switching ranging algorithm based on different orbits requires only one true ranging value simultaneously. This scenario can be regarded as a single-satellite positioning algorithm with certain redundancy. In this case, the requirements for satellite visibility are lower, but the reliability is worse. In the original scenario of the integrated navigation algorithm of INS+2-satellite alternate switching ranging under

the LEO 3-satellite on different orbits, under the premise of guaranteeing a real ranging value, an actual ranging value is required to participate in alternate switching ranging. This situation can be regarded as a dual-satellite positioning algorithm with certain redundancy. In this scenario, the reliability is higher, but the requirements for satellite visibility are also higher.

### 4.3.4. Algorithm Robustness Analysis

To explore the robustness of the 3-satellite alternate handover, we start from the model established in Sections 3.3–3.5 and conduct qualitative and quantitative analysis from the perspectives of multipath interference, NLOS interference, LOS interference, noise interference and satellite orbit disturbance. Our angles are explored separately: (1) the coexistence of multipath, NLOS, LOS and the effect of multimode noise on the algorithm. (2) Analysis of LEO satellite orbit disturbance. Without loss of generality, in all the above cases, we only consider two typical representative cases, which are 5 s with a shorter switching time and 60 s with a longer switching time.

(1) Analysis of algorithm robustness under complex interference

To more realistically simulate the navigation and positioning problem in a complex and challenging environment, we will consider multipath, NLOS, LOS and dual-modal Gaussian noise interference at the same time in this section, considering MLNSR = 0.1, $C_\beta = 1$ and MLNSR = 0.5, $C_\beta = 5$ and MLNSR = 1, $C_\beta = 9$, three complex interference scenarios. The simulation results are shown in Figure 10.

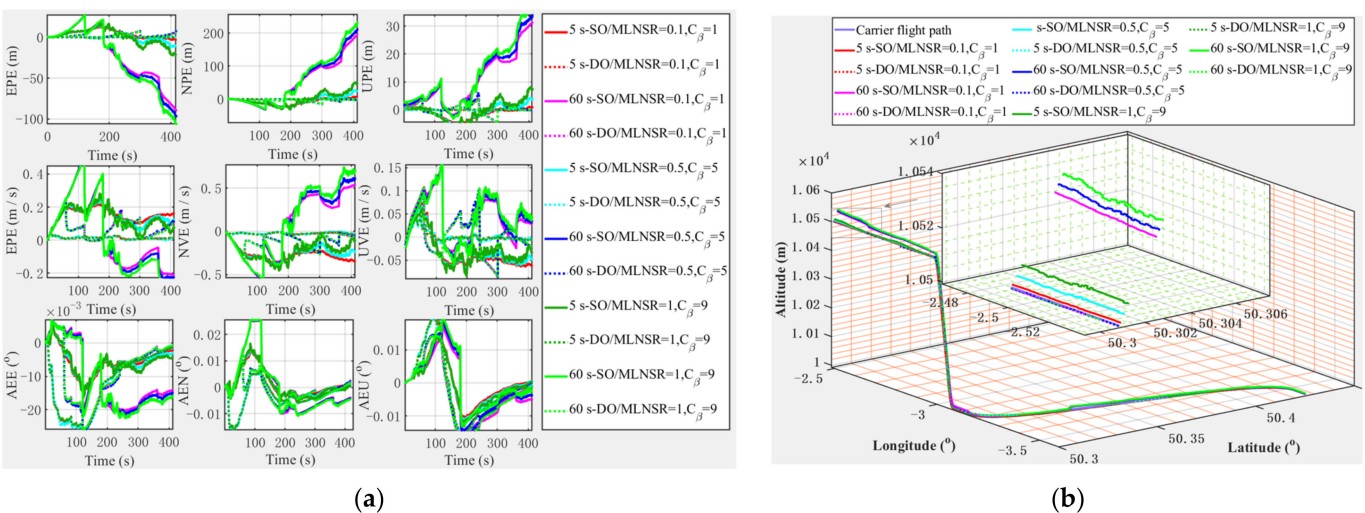

**Figure 10.** Navigation and positioning performance of algorithms under complex interference. (**a**) Error result. (**b**) Trajectory curve.

In Figure 10a,b, we can see that due to the comprehensive interference from multipath, NLOS, LOS and dual-modal Gaussian noise, the positioning result of the algorithm fluctuates and jitter is relatively greater. It is not difficult for us to understand this result and the final positioning data with alternate switching times of 5 s and 60 s are statistically drawn into statistical Figure 11.

In Figure 11a,b, we can see that in this type of complex interference environment, with the increase in MLNSR and dual-modal Gaussian noise intensity, the corresponding positioning error will increase, but the accuracy of the positioning algorithm can still be maintained at the meter level.

(2) Analysis of LEO satellite orbit disturbance

According to the model description in Section 3.5, combined with other relevant simulation parameters, we simulated the comprehensive perturbation factor model of the Earth's nonspherical perturbation and atmospheric drag perturbation. As a comparison, we added the simulation results without perturbation. The simulation results are shown in

Figure 12, where OP represents orbital perturbation and other acronyms have the same meaning as the preceding text.

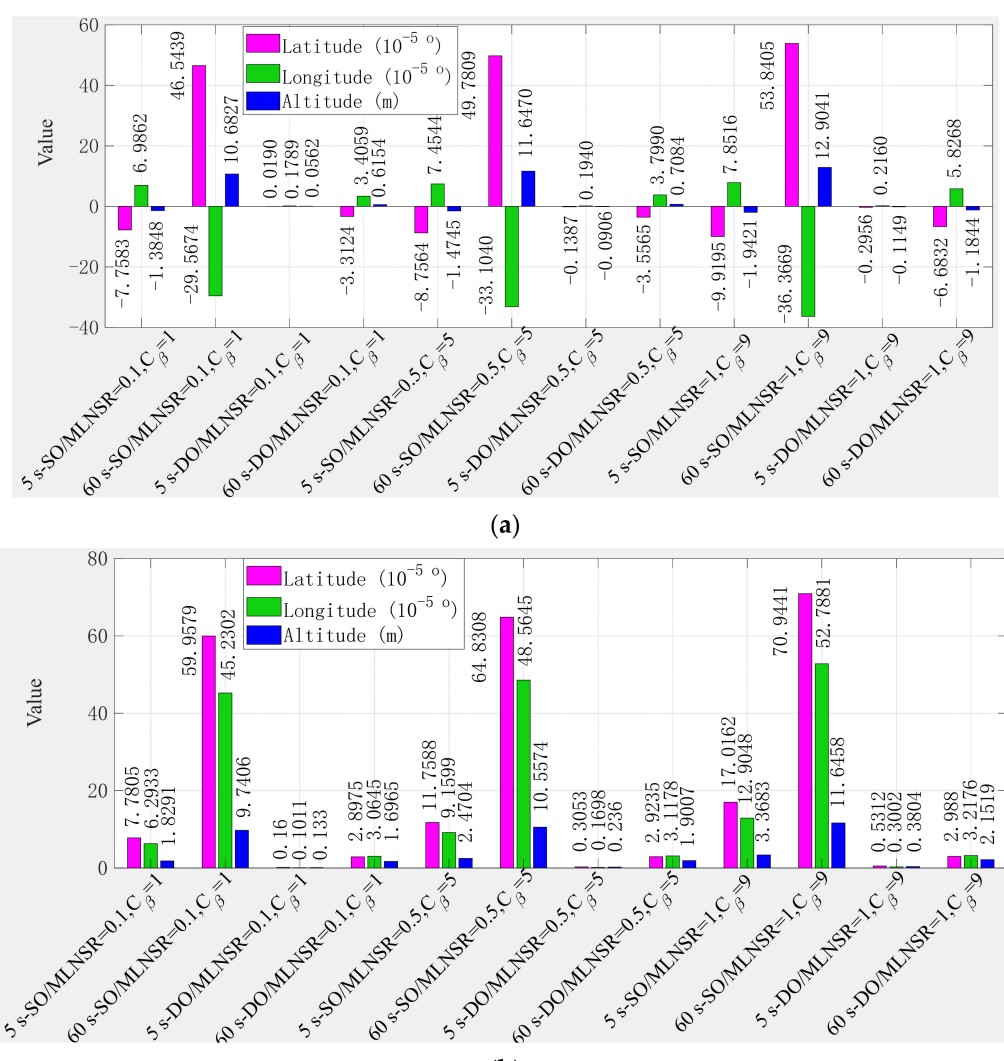

**Figure 11.** Algorithmic navigation and positioning results statistics under complex interference. (**a**) Mean statistics. (**b**) Standard deviation statistics.

In Figure 12a,b, we can intuitively see that under different orbits, when the switching time is short, the orbital perturbation has little effect on navigation and positioning and when the switching time is long, we can see that the influence of orbital perturbation on navigation and positioning increases, but the effect is not particularly obvious. Under the same orbit, even in a short switching time, the result of orbital perturbation on navigation positioning is obvious and under a longer switching time, the impact of orbital perturbation shows a more serious trend. To facilitate quantitative analysis, we show the mean and standard deviation statistics of the corresponding navigation and positioning trajectory, as shown in Figure 13.

In Figure 13a,b, we can see that, in general, the influence of orbital perturbation on the navigation and positioning results makes the algorithm worse than the navigation and positioning results in the case of no orbital perturbation, but there will be fluctuations in the standard deviation; for example, when the switching time is 5 s, the impact of orbital perturbation on different orbits will "improve" the navigation and positioning performance and, similarly, when the switching time is 60 s, the influence of orbit perturbation on the same orbit also makes the navigation and positioning performance "improved". This

phenomenon is also an objective manifestation of the impact of orbital perturbation on navigation and positioning; however, in addition to the local jitter caused by the impact of orbital perturbation, the overall navigation and positioning results are within an acceptable range and this impact can be compensated by upgrading the satellite's orbit in the later stage.

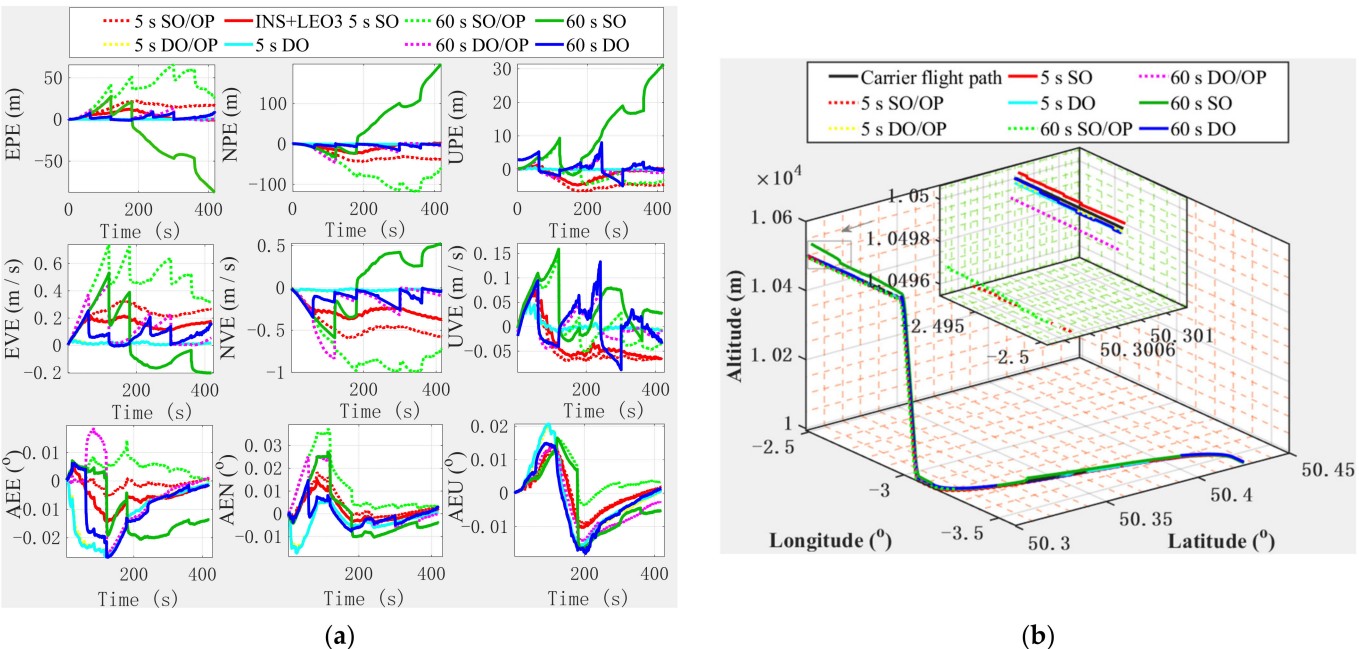

(**a**)　　　　　　　　　　　　　　　　　　　　　　　　(**b**)

**Figure 12.** Navigation and positioning error curve under the combined perturbation of aspherical earth perturbation and atmospheric drag perturbation. (**a**) Error result. (**b**) Trajectory curve.

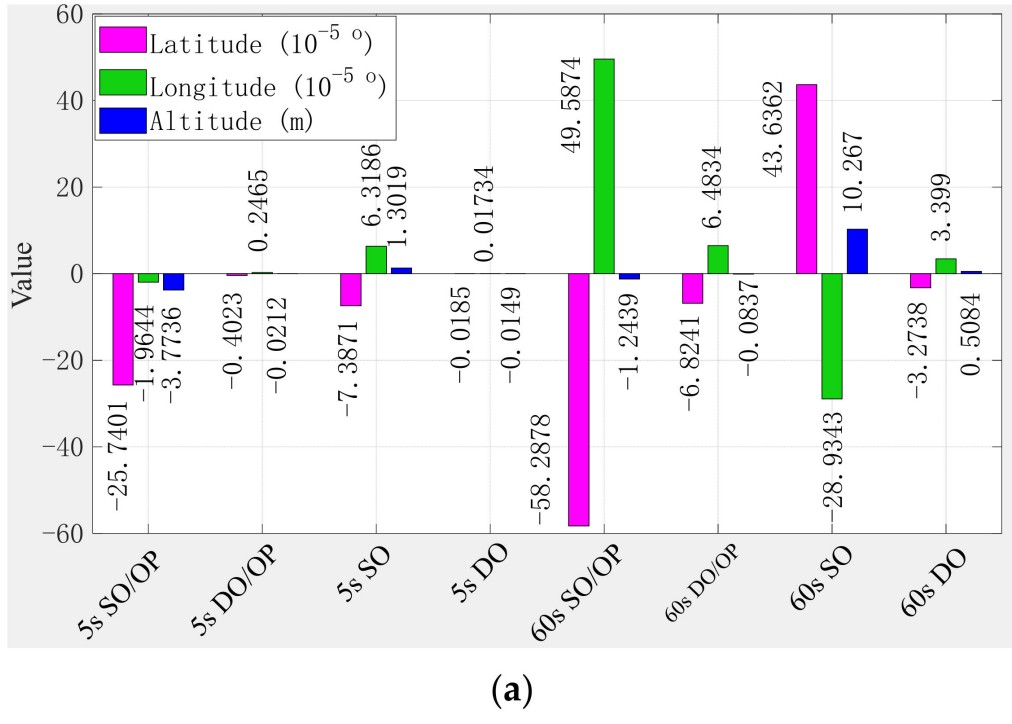

(**a**)

**Figure 13.** *Cont.*

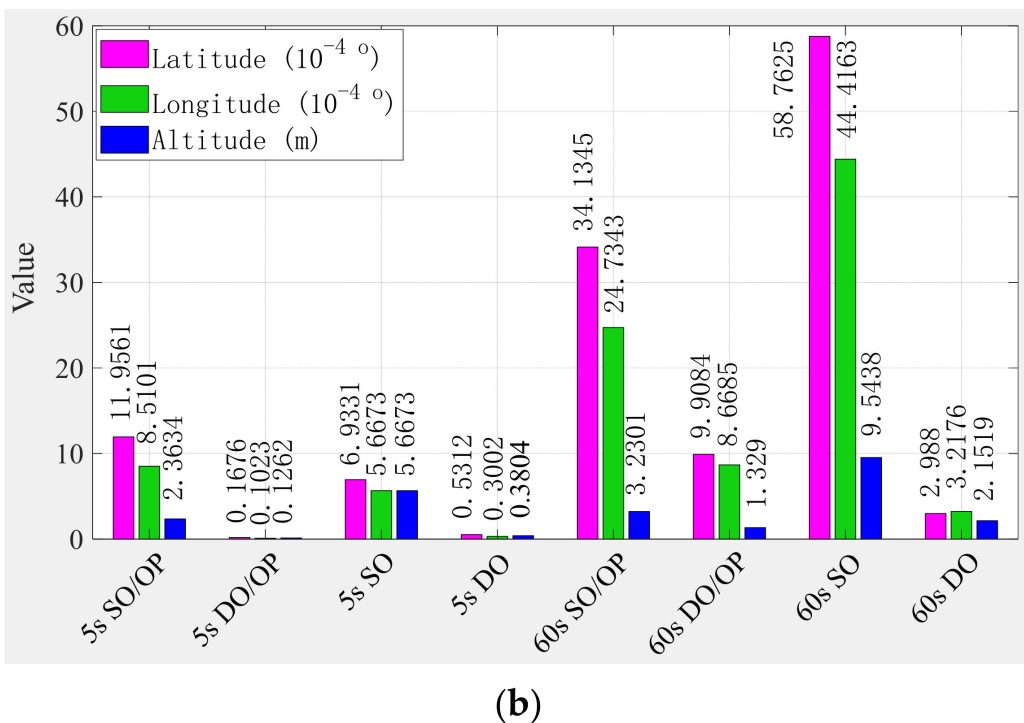

**(b)**

**Figure 13.** Algorithmic navigation and positioning result statistics under the combined perturbation of aspherical earth perturbation and atmospheric drag perturbation. (**a**) Mean statistics. (**b**) Standard deviation statistics.

From the above analyses, it can be seen that the 3-satellite alternate switching algorithm has strong anti-jamming capability and robustness, so it is very suitable for navigation and positioning solutions in complex challenging environments. However, in practical applications, we must also consider the impact of orbital perturbation on the navigation and positioning algorithm and try to suppress the errors caused by the perturbation as much as possible.

## 5. Algorithm Comparison and Analysis

We present the relevant parameters of our algorithm and some of the advanced location service algorithms in Tables 8 and 9 and we compare each indicator, where/means that the corresponding article is not mentioned or not given and 0.00 may be the original author's rounded value, which is not considered. It is marked with 0.00*, where we convert the unit of m in refs [18,19] to rad and the conversion equation is as follows [48]:

$$\begin{cases} \text{Lon } (m) = R_0 \times \text{Lon } (rad) \\ \text{Lat } (m) = R_0 \times \text{Lat } (rad) \end{cases} \tag{25}$$

where $R_0$ represents the radius of the Earth. Then, according to 1 rad = 57.29578°, the radians are converted to deg. Here, we use the average radius of the Earth $R_0 = 6371.393$ km.

Tables 8 and 9 show that our algorithm is significantly better than other algorithms in terms of position, velocity and attitude performance indicators, especially in terms of attitude, longitude, latitude and altitude performance. The attitude error performance is better than the algorithm in [21] by 2 to 3 orders of magnitude. From the final navigation and positioning trajectory error perspective, although the standard deviation in latitude is worse than the algorithms in [18,19], our algorithm's altitude performance is the highest among all algorithms and the mean and standard deviation positioning accuracy reach the cm and dm levels, respectively.

**Table 8.** The position, speed and attitude error statistics of different scene algorithms with a switching time of 5 s.

| Error Index | | EPE (m) | NPE (m) | UPE (m) | EVE (m/s) | NVE (m/s) | UVE (m/s) | AEE × 10⁻³ (°) | AEN × 10⁻³ (°) | AEU × 10⁻³ (°) |
|---|---|---|---|---|---|---|---|---|---|---|
| Mean | DO | 0.123 | 0.020 | 0.142 | 0.012 | −0.020 | 0.001 | −1.257 | −3.385 | −1.062 |
| | OE/DO | 0.115 | −0.245 | −0.068 | 0.011 | −0.019 | 0.0005 | −1.243 | −0.055 | −0.044 |
| | [3] | 22.644 | −8.472 | −150.414 | −6.639 | 0.502 | −3.004 | / | / | / |
| | [18] | / | / | / | −0.008 | 0.013 | 0.006 | / | / | / |
| | [19] | −0.252 | −0.088 | / | 0.001 | 0.001 | −0.002 | / | / | 256 |
| | [20] | −0.206 | 1.103 | / | / | / | / | / | / | 335 |
| | [21] | / | / | / | 0.00* | 0.00* | −0.08 | 670 | 0.00* | 4170 |
| Std | DO | 0.214 | 0.591 | 0.380 | 0.009 | 0.014 | 0.012 | 0.912 | 4.915 | 5.359 |
| | OE/DO | 0.067 | 0.217 | 0.111 | 0.006 | 0.007 | 0.011 | 4.875 | 4.181 | 4.516 |
| | [3] | 3.682 | 4.109 | 39.417 | 0.039 | 0.196 | 0.216 | / | / | / |
| | [18] | / | / | / | 0.66 | 0.501 | 0.157 | / | / | / |
| | [19] | 0.836 | 0.678 | / | 0.043 | 0.028 | 0.118 | / | / | 121 |
| | [20] | 1.317 | 1.730 | / | / | / | / | / | / | 2216 |
| | [21] | / | / | / | 0.12 | 0.12 | 0.10 | 1210 | 260 | 9680 |

**Table 9.** Trajectory error statistics of different scene algorithms under a switching time of 5 s.

| Algorithm | Mean | | | Std | | |
|---|---|---|---|---|---|---|
| | Lon × 10⁻⁶ (°) | Lat × 10⁻⁵ (°) | Alt (m) | Lon × 10⁻⁶ (°) | Lat × 10⁻⁵ (°) | Alt (m) |
| DO | −0.018 | 0.017 | −0.014 | 0.531 | 0.300 | 0.380 |
| OE/DO | −0.222 | 1.615 | −0.068 | 0.195 | 0.094 | 0.111 |
| [3] | 8000 | 10300 | −0.76 | 1115 | 377.000 | 15.118 |
| [18] | −0.207 | −0.018 | 0.15 | 0.791 | 0.044 | 0.281 |
| [19] | / | / | 0.198 | 0.336 | 0.048 | 0.591 |
| [20] | / | / | 0.187 | / | / | 0.423 |
| [21] | / | / | 0.85 | / | / | 2.10 |

From the analysis in Section 5, our algorithm is significantly better than INS and some other advanced algorithms in most performance indicators. The positioning accuracy can reach the dm level and we use a low-cost, low-complexity and anti-jamming design solution. The above results also show that our algorithm has higher accuracy and stronger stability. Therefore, our algorithm can be used as a cm level location service solution in harsh environments, such as lush forests, canyons and high-latitude areas with incomplete visual satellites.

## 6. Conclusions and Future Work

Our aim is to improve the accuracy, reliability and anti-jamming performance of location services in harsh environments. We strive to provide a solution not requiring elevation information in extremely harsh environments, such as forests, canyons, cities and high latitudes, with few visible satellites to improve the accuracy, reliability and anti-jamming performance of location services in harsh environments. We provide cm level navigation and positioning solutions. This paper is based on the technical requirements of ICN, based on the full-duplex mechanism or time synchronization technology of LEO satellites, assuming that the clock error between the LEO satellite and the receiver has been eliminated and gives a reference scheme for using three LEO satellites for high-precision navigation and positioning when the visible satellites are not complete.

By introducing the concept of real ranging values and virtual ranging values, according to the different orbital satellites and switching times used, we conducted a simulation experiment to draw the following conclusions:

(1) Regardless of whether it is on the same orbit or different orbits or the original scenario and the comparison scenario, there is a tendency for the algorithm error to gradually increase as the alternate time increases.

(2) Under the same switching time, the performance of the switching algorithm under different orbits is better than that of the same orbits; the alternate switching algorithm of the original scenario is better than the switching algorithm of the comparison scenario.

(3)     Among these algorithms, the performance of the original scenario based on the integrated navigation algorithm based on INS+2-satellite alternate switching ranging under the LEO 3-satellite is the best, followed by the INS+LEO3-satellite alternate switching ranging algorithm under different orbits. However, each has its advantages and disadvantages and in actual business engineering, we should choose according to the actual situation.

(4)     Compared with conventional MEO constellation navigation systems, such as GPS, GNLONASS, Galilei and BDS, LEO constellations are more suitable for ICN solutions due to their low deployment cost and high navigation and positioning performance.

(5)     For multipath, NLOS and LOS interference, with the increase in MLNSR, the error of the algorithm also increases; for the environment of dual-modal Gaussian noise interference, the increase in the navigation error of the algorithm is also increasing; in the complex interference environment, the algorithm error is relatively large. In addition, in practical applications, since LEO satellites are subject to relatively large orbital perturbations, especially aspherical perturbation of the Earth and atmospheric resistance perturbations, the overall navigation and positioning results are acceptable. However, regardless of the kind of interference situation, our algorithm can guarantee very good robustness and can meet the demand of location services in challenging environments.

Our algorithm is based on an LEO giant internet satellite system. At present, LEO satellites have excellent research prospects and practical significance. Therefore, our algorithm has practical significance. Relying on many LEO satellites, our algorithm can provide a new location reference solution for real-time location services and search and rescue in harsh environments, such as forests, gullies and canyons. In addition, compared with the traditional tightly coupled GNSS/INS integration algorithm, our algorithm not only has navigation and positioning accuracy and strong robustness, but also has strong anti-jamming performance. However, there is room for improvement in positioning accuracy in our algorithm. Thus, providing precise clock deviation-specific elimination technology will be the focus of our future research work.

**Author Contributions:** Y.Y. and L.Y. conceived the conceptualization and algorithm, L.Y. completed the implementation of the algorithm and the writing of the paper and supported the writing—review and editing. N.G. completed some preliminary simulations and did preliminary research and summary. X.L. provided theoretical guidance and suggestions for revision of the paper. All authors have read and agreed to the published version of the manuscript.

**Funding:** This research was supported by the National Key Research and Development Program of China (Grant Nos.2017YFC1500904, 2016YFB0501301; Funder, Yikang Yang), National 973 Program of China (Grant Nos.613237201506; Funder, Yikang Yang), Advance Research Project of Common Technology (No.41418050201; Funder, Haifeng Yang.) and Open Research Fund of Southwest China Institute of Electronic Technology (No.H18019; Funder, Haifeng Yang).

**Institutional Review Board Statement:** Not applicable.

**Informed Consent Statement:** Not applicable.

**Data Availability Statement:** Not applicable.

**Conflicts of Interest:** The authors declare no conflict of interest.

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
