# Peer review of "A High-Precision and Low-Cost Broadband LEO 3-Satellite Alternate Switching Ranging/INS Integrated Navigation and Positioning Algorithm"

_drones, doi:10.3390/drones6090241_

Round 1

Reviewer 2 Report

An integrated navigation algorithm based on LEO satellite communication and navigation integration with 3-satellite alternate switch ranging is proposed to solve the problem of location services in harsh environments. However, there are some points should be improved in this manuscript:

(1) The "basic principle" part is too long, why not present the proposed algorithm directly and show the "basic principle" by citations.

(2) The simulations in Section 4 and Analysis in Section 5 are to redious for the scientific research, authors should present the main contributions as clearly as possible.
